# Multi-Task Time Series Forecasting Based on Graph Neural Networks

**DOI:** 10.3390/e25081136

**Published:** 2023-07-28

**Authors:** Xiao Han, Yongjie Huang, Zhisong Pan, Wei Li, Yahao Hu, Gengyou Lin

**Affiliations:** Command Control Engineering College, Army Engineering University of PLA, Nanjing 210007, China; h.x.good@163.com (X.H.); yongjiehforever@163.com (Y.H.);

**Keywords:** multi-task learning, cross-timestep feature sharing, dynamic dependency, attention mechanism, graph neural network

## Abstract

Accurate time series forecasting is of great importance in real-world scenarios such as health care, transportation, and finance. Because of the tendency, temporal variations, and periodicity of the time series data, there are complex and dynamic dependencies among its underlying features. In time series forecasting tasks, the features learned by a specific task at the current time step (such as predicting mortality) are related to the features of historical timesteps and the features of adjacent timesteps of related tasks (such as predicting fever). Therefore, capturing dynamic dependencies in data is a challenging problem for learning accurate future prediction behavior. To address this challenge, we propose a cross-timestep feature-sharing multi-task time series forecasting model that can capture global and local dynamic dependencies in time series data. Initially, the global dynamic dependencies of features within each task are captured through a self-attention mechanism. Furthermore, an adaptive sparse graph structure is employed to capture the local dynamic dependencies inherent in the data, which can explicitly depict the correlation between features across timesteps and tasks. Lastly, the cross-timestep feature sharing between tasks is achieved through a graph attention mechanism, which strengthens the learning of shared features that are strongly correlated with a single task. It is beneficial for improving the generalization performance of the model. Our experimental results demonstrate that our method is significantly competitive compared to baseline methods.

## 1. Introduction

Time series generally refers to a set of random variables derived from the observation of the development and change process of something and collected at a certain frequency, with characteristics of time dependency, seasonality, trend, and randomness. Time series forecasting is vital in many real-world scenarios, such as traffic forecasts [1,2], air quality prediction [3,4,5], and water quality monitoring [6]. Especially in the field of healthcare, the forecasting of future incidence and mortality rates among patients enables effective control and prevention of diseases [7,8]. However, it is challenging to improve model prediction accuracy by effectively capturing dynamic dependencies in time series data.

Each data point in a time series is influenced by all preceding data points [9], which signifies a prevalent yet significant global dynamic dependency. Disregarding this global dependency can result in substantial bias in current predictions. Particularly when predicting mortality in ICU patients, it is crucial to consider the features of the patients at all timesteps, as they are temporally correlated. Numerous efficient machine learning methods have been developed to capture the global dependency of time series. Traditional machine learning models, such as ARIMA [10], as well as machine learning-based methods such as Neural Networks [11] and SVM [12], are proposed to deal with it, yielding encouraging performance. Nevertheless, these methods may not effectively capture long-term dependencies in the data and treat all historical data as equally important without thoroughly considering dynamic dependencies within the data.

Innovative achievements have been attained with deep time series prediction methods supported by recurrent neural network (RNN) architectures, specifically long and short-term memory (LSTM) [13] units, which exhibit better capabilities in capturing dynamic dependencies within the data. However, its sequential nature hampers parallelized computation, especially when dealing with lengthy sequences. Recent work by Vaswani et al. [14] demonstrates that the attention mechanism proves effective in modeling long-term dependencies in sequences unbound by distance constraints. The attention mechanism assigns weights to historical data based on the current input and has been shown to effectively capture global dependencies between inputs and outputs. Wu et al. [15] propose a dual self-attentive network (DSANet) specifically for dynamic periodic or acyclic multivariate time series forecasting. The temporal fusion Transformer [16] specifically incorporates a temporal self-attention decoder to learn any long-term dependencies present within the dataset, further demonstrating the advantages of the attention mechanism in time sequence forecasting.

To improve the accuracy of prediction tasks, numerous models have incorporated the concept of multi-task learning (MTL) from the domain of machine learning. Multi-task learning aims to learn multiple interconnected tasks together to improve the learning of a model for each task by using the knowledge contained in all or some of the other tasks [17]. For instance, the knowledge gained during the initial phases of heightened infection risk can act as a precursor for later stages marked by elevated mortality risk. Skillfully harnessing this temporal relationship between tasks holds the potential to significantly augment the predictive precision of the model. Harutyunyan et al. [18] demonstrate that the proposed multitask learning architecture allows to extract certain useful information from the input sequence that single-task models could not leverage, which explains the better performance of multitask LSTM in some settings. Durichen [19] explores the potential of a multi-task Gaussian process (MTGP) model for physiological time series analysis, with the objective of learning correlations between and within tasks simultaneously. While these methods have improved the performance of models through knowledge sharing during the process of multi-task learning, the temporal dynamics of shared features and the strong correlation between tasks have not been considered.

In the context of time series forecasting tasks, acquiring knowledge from a specific timestep for one task (e.g., predicting sepsis onset) may facilitate learning for another task at a later timestep, such as predicting mortality rate. However, the relevance of the acquired shared knowledge to the tasks depends on the similarity of these cross-timestep features between tasks, and this similarity also varies dynamically over time (as a form of local temporal dependency). Consequently, designing sharing strategies that enable models to effectively capture the local dynamic dependency of shared features across timesteps between tasks while strengthening their strong correlation with each individual task poses a significant challenge.

Generally, graph neural networks (GNNs) [20] assume that the state of a node is influenced by the states of its neighboring nodes. They can enhance learning accuracy and speed by leveraging both spatial and temporal information on the graph structure. Consequently, capturing the spatiotemporal dynamic dependencies among time series features using GNNs has gained significant attention in recent years in the field of time series forecasting research [21]. GNNs can be categorized into two types based on the aggregation of node features: Graph Convolutional Networks (GCNs) [22] and Graph Attention Networks (GATs) [23]. GCN-based methods rely on knowing the topology in advance and assigning equal weights to neighbors during aggregation. However, they do not fully consider the dynamic changes in dependencies between features.

In contrast, GATs introduce a self-attentive mechanism that can dynamically capture the correlations of spatial features and adaptively learn the weights of each node over time. This enhances the inductive learning capability of graph models. STNN [24] utilizes GATs to model complex and dynamic spatial correlations without requiring expensive matrix operations or relying on predefined road network topologies. On the other hand, You [25] captures dynamic spatial correlations through spatial attention networks but only considers the correlation between spatial features at the same timestep, neglecting the influence of spatial features at earlier timesteps. Taking inspiration from GATs, we propose a strategy for sharing cross-timestep features, where potential features from other tasks at the same or different timesteps are aggregated through attention-based fusion to achieve feature sharing. In our approach, the edges representing the inter-task feature correlations are initially unknown and are learned based on the similarity between cross-timestep features to capture strongly correlated shared features.

To summarize our approach’s motivation: (1) The attention mechanism is capable of capturing long-term dependencies in time series data. Hence, we leverage this mechanism to capture global dynamic dependencies between different timestep features within each task. (2) In time series prediction tasks, learning a specific timestep feature from a related task (e.g., disease onset) can be beneficial for predicting a later timestep feature from another specific task (e.g., predicting death). Leveraging the powerful representation capabilities of graph structures in capturing complex nonlinear correlations, we learn a graph structure based on the feature correlations across different timesteps and tasks to model the local dynamic dependencies in the data. (3) GAT is particularly effective in handling dynamic and non-uniform graph features by adaptively determining the input weights of each node using an attention mechanism. Therefore, we propose a sharing strategy to effectively capture the dynamic dependencies among features across timesteps in time series prediction and enhance the accuracy of the model predictions.

In the G-MTL framework, as illustrated in Figure 1, we first employ an attention mechanism to capture the global dynamic dependencies among potential features at different timesteps within each task. Subsequently, we identify strongly correlated cross-timestep features across tasks and learn a graph structure to capture the local dynamic dependencies of shared features. Finally, we utilize a graph attention network to aggregate the cross-timestep features across tasks for enhanced feature sharing. Our experimental results demonstrate the competitiveness of our approach compared to the current state-of-the-art methods.

The main contributions of this paper can be summarized as follows:(1)The present paper introduces G-MTL, a novel method for multi-task time series prediction. G-MTL employs the self-attention mechanism to capture the global dynamic dependencies of task-specific features and utilizes the graph attention mechanism to capture the local dynamic dependencies of inter-task features. This approach effectively captures the temporal dynamics inherent in time series data, thereby improving the accuracy of time series forecasting.(2)G-MTL presents an adaptive cross-timestep feature-sharing strategy. It incorporates GATs in a flexible manner to adaptively weigh and aggregate the node features of each task with those of its neighboring nodes. This strategy updates the features of each task at every timestep, facilitating feature sharing across tasks. Throughout the learning process, this sharing strategy acquires shared features that exhibit strong correlations with each task, thereby further augmenting the model’s generalization ability;(3)A series of experiments were conducted on three real multivariate time series datasets to evaluate the predictive performance of G-MTL in comparison to several state-of-the-art time series forecasting models. The results consistently demonstrated that G-MTL outperformed the other models in terms of predictive accuracy.

The remaining sections of this paper are structured as follows. Section 2 provides a brief overview of the existing research literature on the problem of time series forecasting. Section 3 presents a detailed description of the G-MTL model proposed in this paper. To demonstrate the efficacy of our approach, Section 4 presents experimental results obtained from multiple public benchmark datasets and compares them against baseline methods. Finally, in Section 5, we summarize the key findings of this paper and propose future research directions.

## 2. Related Work

### 2.1. Multi-Task Time Series Forecasting

Multi-Task Learning (MTL) has emerged as a promising approach for enhancing model representation and generalization by simultaneously learning multiple tasks. Historically, multi-task learning models have focused on what to share, as the jointly learned models could share instances, parameters, or features. Existing MTL studies have predominantly focused on two approaches: parameter-based and feature-based [17].

(1) Parameter-based MTL leverages model parameters, such as coefficients in a linear model or weights in a deep model, from one task to assist in learning model parameters for other tasks. This is commonly achieved through techniques such as regularization. For example, MasterGNN [26] introduces a multitask adaptive training strategy that automatically balances the multitasking adversarial learning discriminant loss to improve overall accuracy. DeepTTE [27] learns a multitasking loss function by considering the trade-off between individual and collective estimation. Meanwhile, MultiTL-KELM [28] employs a multi-task learning algorithm to transfer knowledge learned from previous data to the current prediction task for multi-step ahead time series prediction. These models exploit the sharing of model parameters among tasks to mitigate overfitting risks during training but heavily rely on prior task relationships as a source of information.

(2) Feature-based MTL approaches aim to extract common features across tasks to facilitate knowledge sharing. Given the interrelated nature of tasks, it is reasonable to assume that different tasks share certain common features derived from the original feature set. AECRNN [29] enhances the learning of robust features for the master task by incorporating auto-encoders into each time series, thereby creating a multitask learning framework. MTL-Trans [30], based on transformers, introduces two attention-sharing architectures for sharing self-attention features among different tasks. TP-AMTL [31] enables feature transfer within and between tasks across different timesteps, leveraging feature-level uncertainty. While these feature-based MTL methods effectively learn task-general features, they can be vulnerable to the presence of outlier tasks, leading to a degradation in performance. Hence, when enhancing prediction accuracy through multi-task learning feature sharing, it is crucial to consider the robust correlation between the shared features and the specific task at hand.

### 2.2. Graph Neural Networks (GNNs)

Graph Neural Networks (GNNs) offer a promising approach for modeling multivariate time series data, thanks to their ability to handle arrangement invariance, local connectivity, and combinatorics [32]. This methodology leverages the correlations among time series while preserving their temporal trajectories, leading to enhanced prediction accuracy in time series forecasting tasks. Recent studies have demonstrated the effectiveness of GNNs in addressing relational dependencies. GCNs aggregate the features of neighboring nodes to model the feature nodes efficiently. In the context of time series data, specialized architectures such as GCRN [33], STGCN [34], and T-GCN [35] have been developed, combining recurrent units with GCNs. These architectures have exhibited promising results in various time series forecasting tasks, demonstrating how GNNs effectively capture both the temporal and spatial dependencies of time series data.

Although GCNs excel in learning complex topological structures to capture spatial dependence, they have a limitation in that they aggregate neighbor nodes without considering the varying importance of different neighbors. GATs introduced a self-attention mechanism that assigns different weights to each node in the graph based on its neighbor node features while computing its representation. For instance, frameworks such as GAT-LSTM [36], TC-GATN [37], and GCAR [38] utilize attention graph networks to model intricate and dynamic spatial correlations, leading to improved prediction performance. The advantage of GATs lies in their ability to train without requiring knowledge of the entire graph structure; only the neighboring nodes of each node are considered, leading to parallel computation on different nodes and fast computation speed. GATs adaptively capture the correlation between the current node and its neighboring nodes based on their features. Inspired by this, we model the graph structure of different timesteps in the model and then dynamically capture the local dynamic dependence of the feature through the self-attention mechanism.

### 2.3. Attention Mechanism

In time series prediction, one direction of deep learning models is to capture temporal patterns in dynamic time series data [39], and recurrent neural network (RNN)-based prediction models have started to gain popularity, such as DeepAR [40]. Although LSTMs overcome the problem of gradient disappearance or explosion to some extent, RNN-based models still do not model long-term dependencies well [13].

Attention mechanisms have emerged as a highly successful approach in the field of deep learning, particularly for tackling sequence-to-sequence problems. The goal of attention is to prioritize and focus on the most relevant information for the current task while reducing attention to other less significant information. Within academic research, there exist various variants of attention mechanisms, including soft attention, self-attention, and Transformer.

Soft attention is a widely used type of attention mechanism. For instance, the two-stage attention mechanism-based RNN (DARNN) proposed in [41] is designed to capture long-term time-dependent relationships and select pertinent driving sequences for prediction. Another model called MARNN [42], based on the attention mechanism, not only captures dependencies and weights within the driving sequence but also incorporates temporal correlations across timesteps.

Self-attention, a variant of soft attention, allows for associating different positions within a single sequence to calculate the representation of that sequence. In the [43], a spatial self-attention structure is developed to capture spatial information of high-dimensional variables, while a temporal self-attention structure captures the temporal evolution of the target variable. The LSTM-based model SAnD [44] employs a masked self-attentive mechanism to overcome the limitation of sequential processing and improve efficiency when dealing with long sequences.

Among all the attention-based variants, the Transformer model has emerged as one of the most efficient paradigms for handling long-term sequence modeling. It introduces several enhancements to soft attention [45], enabling sequence-to-sequence modeling without the need for recurrent network units. For instance, the MTL-Trans model [30] employs Transformer to capture long-term temporal dependencies between shared and private features separately. At the same time, TFT [16] utilizes a self-attentive mechanism to learn long-term dependencies across different timesteps. An improved Transformer model called Informer is proposed in [46], which incorporates the ProbSparse self-attentive mechanism to address the challenges of high time complexity and memory consumption associated with the Transformer model. This enhancement significantly improves the efficiency of long-time series prediction. Additionally, [47] introduces the LogSparse Transformer, specifically designed to alleviate the memory bottleneck encountered in long sequence modeling.

## 3. Our Method

In this section, we propose the G-MLT model for multi-task time series forecasting, which incorporates feature sharing across timesteps. The model framework is illustrated in Figure 1. We initiate the discussion with the problem formulation in Section 3.1. Subsequently, in Section 3.2, we elucidate how the self-attention mechanism effectively captures the global dynamic dependency of intra-task features. Furthermore, in Section 3.3, we leverage Graph Attention Networks (GATs) to capture the local dynamic dependencies of inter-task temporal steps and adaptively share features. Finally, we define the objective function for optimization in Section 3.4.

### 3.1. Problem Formulation

We are given *M* tasks ${\left\{T_{i}\right\}}_{i=1}^{M}$, and corresponding multivariate time series training dataset $D_{i}=\left\{(X^{i},y^{i})\right\}={\left\{x_{k}^{i},y_{k}^{i}\right\}}_{k=1}^{N_{i}}$ with $N_{i}$ training samples, $x_{k}^{i}\in R^{T\times d}$ and its associated label $y_{k}^{i}\in \left\{1,2,\cdots ,c_{i}\right\}$, where $c_{i}$ is the number of classes in the dataset $D_{i}$. Assuming that there are *M* task networks ${\left\{N_{T_{i}}\right\}}_{i=1}^{M}$ and *L* layers of feedforward neural networks for each task, where the input feature map of task $T_{i}$ in the l(1≤l≤L) layer is denoted as $F_{T_{i}}^{l-1}=\left\{f_{T_{i_{1}}}^{l-1},f_{T_{i_{2}}}^{l-1},\cdots ,f_{T_{i_{T}}}^{l-1}\right\}$, and $f_{T_{i_{t}}}^{l-1}\in R^{d}$ is the timestep *t* input feature. The corresponding output is $F_{T_{i}}^{l}\in R^{T\times d}$. The proposed model is discussed by taking any two tasks $T_{i},T_{j}$(i≠j) as examples. In single-task learning, $N_{T_{i}}$ is used to make predictions for the task $T_{i}$. In a multi-task learning framework, we improve task generalization performance by sharing useful features learned from other tasks. Specifically, the features learned from other tasks $F_{T_{j}}^{l}$(∀j≠i) are used with the features $F_{T_{i}^{l}}$ of task $T_{i}$ to construct the graph structure *A* and use it to assist task $T_{i}$ in making predictions.

### 3.2. Intra-Task Feature Global Dynamic Dependencies

In time series data, the features of each timestep undergo changes, and the degree of correlation also varies accordingly. To capture this dynamic dependence, we leverage a self-attention mechanism to capture the inter-dependency of features across the different timesteps involved in the task. The specific process is depicted in Figure 2.

For any given task $T_{i},$$i\in \left\{1,2,\cdots M\right\}$, perform a linear transformation of the input feature $F_{T_{i}}^{l-1}$ with Q,K,V:(1)$$\begin{array}{c} Q_{T_{i}}^{l}=Q\left(F_{T_{i}}^{l-1}\right) \end{array}$$(2)$$\begin{array}{c} K_{T_{i}}^{l}=K\left(F_{T_{i}}^{l-1}\right) \end{array}$$(3)$$\begin{array}{c} V_{T_{i}}^{l}=V\left(F_{T_{i}}^{l-1}\right) \end{array}$$
where $Q_{T_{i}}^{l},K_{T_{i}}^{l},V_{T_{i}}^{l}\in R^{T\times d}$ represent the output of the feature map of task $T_{i}$ after linear transformation, and then $Q_{T_{i}}^{l},K_{T_{i}}^{l}$ are subjected to a multiplication operation of the matrix to calculate the correlation weights between different timesteps and scored with a softmax operation. Finally, the attention feature vectors are calculated by weighted summation:(4)$$F_{T_{i}}^{l}=softmax\left(Q_{T_{i}}^{l}{\left(K_{T_{i}}^{l}\right)}^{T}\right)V_{T_{i}}^{l}$$
where $F_{T_{i}}^{l}$ represents the feature matrix of task $T_{i}$ after attention. To compensate for the information lost when the attention mechanism captures feature correlation, we fuse the attention feature vector $F_{T_{i}}^{l}$ with the input feature vector $F_{T_{i}}^{l-1}$, calculated as follows:(5)$$C_{T_{i}}^{l}=F_{T_{i}}^{l-1}+F_{T_{i}}^{l}$$
where $C_{T_{i}}^{l}$ represents the fusion feature of the output, and the features on each timestep are $C_{T_{i_{1}}}^{l},C_{T_{i_{2}}}^{l},\cdots ,C_{T_{i_{T}}}^{l}$, + represents the addition of the corresponding elements of the two matrices. We use $C_{T_{i_{1}}}^{l},C_{T_{i_{2}}}^{l},\cdots ,C_{T_{i_{T}}}^{l}$ as the input features of the following graph structure nodes.

### 3.3. Inter-Task Feature Local Dynamic Dependencies

The model has the important objective of capturing local dynamic feature dependencies across timesteps between tasks. To achieve this, we employ a directed graph structure, where the nodes represent timesteps, and the edges represent the feature dependencies between them. However, we establish edges only between timesteps that exhibit strong correlations. Through the utilization of the graph attention mechanism, we learn shared features that exhibit strong correlations with each task.

#### 3.3.1. Graph Structure Learning

To effectively model the complex and nonlinear correlations between tasks, which involve multiple cross-timestep features, we learn a graph structure that represents the nodes as timesteps and encodes their correlations as the edges in the graph. Considering the asymmetrical back-and-forth dependency between time series timesteps, we utilize a directed graph to accurately represent this pattern of dependency. The edges from node *s* to node *t*(s<t) indicate that the timestep *s* is utilized to model the dependencies on timestep *t*. An adjacency matrix *A* is employed to encode this directed graph, where $A_{s,t}^{l}$ represents the presence of directed edges from node *s* to node *t* in layer *l*. Refer to Figure 3 for a visualization of the graph learning process.

Since the learned features for different tasks may have distinct representations, constructing a graph structure between tasks may not be optimal. To address this, the task-specific features are transformed into a shared potential space using an additional network *G*. For each task $T_{i}$(i=1,2,⋯,M), this transformation is denoted as:(6)$${\tilde{C}}_{T_{i_{t}}}^{l}=G\left(C_{T_{i_{t}}}^{l}\right)$$
where G(·) is a linear function and ${\tilde{C}}_{T_{i_{t}}}^{l}\in R^{d}$.

When learning the structure of the graph neural network on layer *l*, firstly, the correlations of cross-timestep features between different tasks are calculated. For tasks $T_{i}$ and $T_{j}$:(7)$$\mu_{T_{j_{s}},T_{i_{t}}}^{l}=\frac{{\tilde{C}}_{T_{j_{s}}}^{l}\cdot {\tilde{C}}_{T_{i_{t}}}^{l}}{∥{\tilde{C}}_{T_{j_{s}}}^{l}∥∥{\tilde{C}}_{T_{i_{t}}}^{l}∥},i\neq j$$
$\mu_{T_{j_{s}},T_{i_{t}}}$ reflects the dependency of the timestep *t* feature of task $T_{i}$ on the timestep *s* feature of task $T_{j}$.

Furthermore, considering that there exists a certain level of dependency between the current timestep features of each task and only the same or earlier timesteps of another task, our model explores the similarity between the first *t* timestep features of task $T_{i}$ and all timestep features before the first *s* timestep of the task $T_{j}$.

From these comparisons, the strongly correlated timestep features are selected to construct the graph structure.
(8)$$A_{T_{j_{s}},T_{i_{t}}}^{l}=I\left\{j_{s}\in TopK\left(\left\{\mu_{T_{j_{s}},T_{i_{t}}}^{l}:s\leq t\right\}\right)\right\}$$
Specifically, the TopK strongly correlated edges are chosen, where TopK represents the indices of the first *K* values. This process allows the graph neural network $A^{l}$ to be learned at layer *l*. Subsequently, the graph attention mechanism is employed to facilitate feature sharing between different tasks on the graph neural network $A^{l}$.

#### 3.3.2. Cross-Timestep Feature-Sharing

In our multi-task learning model, we propose a cross-timestep feature-sharing strategy to enable feature sharing among tasks and obtain shared features that are highly relevant to the tasks during the learning process. Through the graph attention mechanism, features across timesteps between tasks are aggregated and shared.

As depicted in Figure 4, in order to transfer knowledge from task $T_{j}$ to time-dependent task $T_{i}$, we enable the feature ${\tilde{C}}_{T_{i_{t}}}^{l}$ of task $T_{i}$ at timestep *t* to gather strongly correlated information from previous timesteps of task $T_{j}$, we then aggregate these correlations into a new feature $Z_{T_{i_{t}}}^{l}$:(9)$$Z_{T_{i_{t}}}^{l}=\alpha_{T_{i_{t}},T_{i_{t}}}^{l}W{\tilde{C}}_{T_{i_{t}}}^{l}+\sum_{j\neq i}^{M}\sum_{A_{T_{j_{s}},T_{i_{t}}}^{l}>0}\alpha_{T_{j_{s}},T_{i_{t}}}^{l}W{\tilde{C}}_{T_{j_{s}}}^{l}$$
Here, ${\tilde{C}}_{T_{j_{s}}}^{l}$ represents the input feature of timestep *s* in the task $T_{j}$, and $W\in R^{d^{\prime }\times d}$ denotes a trainable shared weight that linearly transforms each node feature into a higher-level representation for enhanced expressiveness. It is worth noting that the coefficients are calculated according to the following process:(10)$$\beta_{T_{j_{s}},T_{i_{t}}}^{l}=\sigma \left(a^{T}\left(W{\tilde{C}}_{T_{i_{t}}}^{l}\left|\right|W{\tilde{C}}_{T_{j_{s}}}^{l}\right)\right)$$
where $\beta_{T_{j_{s}},T_{i_{t}}}$ denotes the importance of the node *s* to the node *t*. || denotes concatenate operation, which yields a vector of length $2d^{\prime }$. $a\in R^{2d^{\prime }}$ is a vector of learnable coefficients. $\cdot^{T}$ denotes transpose, and σ is the nonlinear activation function LeakyReLU used to compute the attention coefficients $\beta_{T_{j_{s}},T_{i_{t}}}$.

To make the coefficients easily comparable across nodes, we used the softmax function to score attention to obtain the final attention coefficient $\alpha_{T_{j_{s}},T_{i_{t}}}$:(11)$$\alpha_{T_{j_{s}},T_{i_{t}}}^{l}=\frac{exp\left(\beta_{T_{j_{s}},T_{i_{t}}}^{l}\right)}{\sum_{A_{T_{j_{k}},T_{i_{t}}}^{l}}\left(exp\left(\beta_{T_{j_{k}},T_{i_{t}}}^{l}\right)\right)}$$
$\alpha_{T_{j_{s}},T_{i_{t}}}$ indicates the importance of node *s* to node *t*, i.e., the importance of the first *s* timestep feature of task $T_{j}$ to the first *t*(t>s) timestep feature of task $T_{i}$.

Then, in task $T_{i}$ layer *l*, the output feature for each timestep is:(12)$$f_{T_{i}}^{l}=\sigma \left(Z_{T_{i}}^{l}\right)$$
σ is a nonlinear activation function. Then the graph neural network on the *l* layer output feature is:(13)$$F_{T_{i}}^{l}=\left\{f_{T_{i_{1}}}^{l},f_{T_{i_{2}}}^{l},\cdots ,f_{T_{i_{T}}}^{l}\right\}$$

After the network propagates forward *L* layers, the task $T_{i}$ the output feature matrix is:(14)$$F_{T_{i}}^{L}=\left\{f_{T_{i_{1}}}^{L},f_{T_{i_{2}}}^{L},\cdots ,f_{T_{i_{T}}}^{L}\right\}.$$
Concatenate the results of all nodes as inputs to the fully connected layer for prediction.

For task $T_{i}$, the predicted value of the *k*-th input instance is:(15)$${\hat{y}}_{k}^{i}=f_{\theta }\left(\left[f_{T_{i_{1}}}^{L},f_{T_{i_{2}}}^{L},\cdots ,f_{T_{i_{T}}}^{L}\right]\right)$$

### 3.4. Objective Function

In the G-MTL network, the objective function for task $T_{i}$ can be formulated as the cross-entropy loss:(16)$$L_{T_{i}}=-\sum_{k=1}^{N_{i}}y_{k}^{i}\left(log{\hat{y}}_{k}^{i}\right)$$
Finally, we define the total objective function of the whole network as:(17)$$L_{Total}=\sum_{i=1}^{M}\lambda_{i}\sum_{k=1}^{N_{i}}L_{T_{i}}\left({\hat{y}}_{k}^{i},y_{k}^{i}\right)$$
(·) and $\lambda_{i}$ are the losses to be weighted and their balancing factors, respectively.

## 4. Experiment

In this section, we present the results obtained from several datasets to validate the effectiveness of the proposed method. For our analysis, we defined clinical risks as the occurrence of events (such as Heart Failure, Respiratory Failure, Infection, or Mortality) that may contribute to the deterioration of a patient’s health condition within a specific time window, typically 48 h [31].

### 4.1. Datasets and Tasks

Accurate forecasting of time series data is of significant importance, especially in predicting mortality risk in clinical risk forecasting. Although our method is applicable to various time series forecasting tasks, this paper primarily focuses on clinical time series analysis. Several benchmark datasets in the clinical domain, such as MIMIC-III [48], and PhysioNet [49], have been published and made publicly available. Furthermore, recent studies have proposed clinical prediction benchmarks accompanied by openly accessible datasets [18,50,51,52,53].

MIMIC-III [48] (‘Medical Information Mart for Intensive Care’) is a comprehensive database from a large tertiary care hospital that contains information pertaining to patients admitted to critical care units. The dataset includes vital signs, medications, laboratory measurements, observations and care provider notes, fluid balance, procedure codes, diagnostic codes, imaging reports, hospital length of stay, survival data, and more. It serves as a valuable resource for academic research, industrial applications, quality improvement initiatives, and higher education coursework. The publicly available PhysioNet Challenge 2012 dataset [49] consists of de-identified records from 8000 Intensive Care Unit (ICU) patients. Each record comprises approximately 48 h of multi-variate time series data, encompassing up to 37 features recorded at different time points during the patient’s hospital stay. Examples of these features include respiratory rate, blood glucose levels, and others.

We used three multivariate time series datasets publicly available in the literature [31], which compile for clinical risk prediction from the two open-source EHR datasets. Every dataset used in this paper contains tasks with clear temporal dependencies between them. We briefly describe all the used datasets as follows.

**(1) MIMIC-III Infection**. A collection of 1921 records of patients over the age of 15 admitted to the ICU, where hourly samples were used to construct 48 timesteps from the first 48 h of admission. We selected 12 infection-related variables for the features at each timestep, including Heart Rate (HR), Systolic/Diastolic Blood Pressure (SBP/DBP), Intubation /Unplanned Extubation, Albumin, etc.

**(2) PhysioNet**. A total of 4000 ICU admission records were included, each containing 48 h of records (sampled hourly) and 29 infection-related variables for the features available at each timestep. The features Systolic arterial blood pressure (SBP), Diastolic arterial blood pressure (DBP), Body Temperature (BT), Fractional inspired Oxygen (FiO_2_), and others were used for comparing the performance of the models in our study.

**(3) MIMIC-III Heart Failure**. Hourly sampled data from heart failure patients, a total of 3557 data points with a sufficient amount of features were selected, which included 16 variables at each timestep. In our study, we have selected heart failure-related variables such as Hemoglobin (Hb), Red Blood Cells (RBC), White Blood Cells (WBC), and Platelets for our model selection.

Table 1 summarizes all the experimental data, along with the corresponding statistics for each dataset. After preprocessing, each dataset was divided into a training set and a test set. The model was then trained using all the data up to a specific timestep as the training set, while the test set consisted of data after the last time point seen in the training set.

To thoroughly validate the efficacy of the proposed method, we conducted experiments on various tasks, each with detailed settings as follows.

**Exp 1** Tasks considered for the MIMIC-III Infection dataset were the clinical events before and after infection, Fever (Task 1) as the sign of infection with elevated body temperature, Infection (Task 2) as the confirmation of infection by the result of microbiology tests, and finally, Mortality (Task 3) as a possible outcome of infection.

**Exp 2** Task used in the PhysioNet dataset includes four binary classification tasks, namely, (1) Stay < 3: whether the patient would stay in ICU for less than three days, (2) Cardiac: whether the patient is recovering from cardiac surgery, (3) Recovery: whether the patient is staying in Surgical ICU to recover from surgery, and (4) Mortality prediction (Mortality).

**Exp 3** MIMIC-III Heart Failure contains four different tasks, i.e., (1) Ischemic: an ischemic heart disease, (2) Valvular: a valvular heart disease refers to the abnormal function of the heart valves, (3) Heart Failure, and (4) Mortality.

### 4.2. Experimental Settings

For each experimental trial, we employed cross-entropy as the target function for training. We utilized the Adam optimizer with a learning rate of 0.001 for joint training over 600 epochs. The batch size for the input dataset was set to 256, and we employed a 2-layer LSTM network to extract task-specific features. The sparsity metric Topk, used for graph structure learning, was set at 20%.

### 4.3. Baselines

To assess the effectiveness of our model, we compared it with both single-task and multi-task learning baselines. Multi-task learning (**MTL**) involves simultaneously acquiring knowledge across multiple tasks, enhancing the model’s generalization capabilities through information sharing. In contrast, single-task learning (**STL**) entails the design of a dedicated network for each individual task, with each task being learned independently.**Single-task learning (STL) baselines:****(1) STL-LSTM**, **(2) STL-Transformer**, **(3) STL-RETAIN** [54], **(4) STL-UA** [55], **(5) STL-SAnD** [54], **(6) STL-AdaCare** [56].**Multi-task learning baselines:****(7) MTL-LSTM**, **(8) MTL-Transformer (9) MTL-RETAIN**, **(10) MTL-UA**, **(11) MTL-SAnD**, **(12) AdaCare**, **(13) AMTL-LSTM** [57], **(14) TP-AMTL** [31].

Multi-task learning (MTL) setting with (7) MTL-LSTM, (8) MTL-Transformer [14], (9) RETAIN [54], (10) UA [55], (11) SAnD [54], (12) AdaCare [56] as the base network, respectively.

### 4.4. Comparison Results

We conducted an evaluation of the baseline single-task learning (STL) and multi-task learning (MTL) models, as well as our proposed model, to assess their prediction accuracy on three clinical time series datasets. The evaluation was performed by measuring the Area Under the ROC curve (AUROC). The test accuracy of each comparison method is reported in Table 2, Table 3 and Table 4. From the experimental results, several observations can be made:Overall, most MTL methods outperform STL methods, demonstrating the effectiveness of joint learning of multiple tasks by exploring the relationship between them. Notable examples of such methods include RETAIN and UA.However, it is worth noting that MTL models perform relatively poorly on the MIMIC-III infection dataset, which exhibits clear temporal relationships between tasks. In the MTL-Transformer model, the accuracy achieved in each task using the multi-task learning approach is comparatively lower than that attained by the corresponding single-task models. Additionally, AdaCare with dilated convolution displays severely degraded performance, except for one task. Therefore, it is important to consider not only the temporal dependency of intra-task features but also the temporal dependency of inter-task features. Our proposed model, G-MTL, addresses this concern and leads to significant improvements.We have observed that while MTL models outperform STL models on certain tasks, they suffer from performance degradation on others. This can be clearly observed from the task ranking presented in Table 2, Table 3 and Table 4. The self-attention-based model, SAnD, demonstrates impressive performance on some tasks in the PhysioNet dataset. However, it also experiences performance degradation when used in an MTL setting, resulting in lower overall performance. Although our proposed model, G-MTL, improves model performance, this phenomenon persists. We attribute this to the imbalance of tasks in our setting, leading to unique loss scales for each task.In contrast, our model outperforms the majority of MTL models on all three datasets in terms of performance. This superiority can be attributed to the incorporation of feature sharing across timesteps, which effectively enhances the acquisition of shared knowledge. Consequently, our model exhibits improved task generalization performance by establishing strong correlations with each task. It is worth noting that the G-MTL model surpasses the remarkable TP-AMTL [31] model in the majority of tasks. Under our cross-timestep sharing strategy, which considers the correlation of both intra-task and inter-task features, the model demonstrates superior overall performance.

### 4.5. Analysis of the Interpretability and Effectiveness of the Sharing Strategy

In this section, we aim to provide a comprehensive explanation and validation of the efficacy of the proposed strategy for sharing features across different timesteps.

The tasks in our selected datasets, namely MIMIC-III Infection, PhysioNet, and MIMIC-III Heart Failure, exhibit evident temporal dependencies. Upon examining Table 2, Table 3 and Table 4, it becomes apparent that certain multi-task learning (MTL) models demonstrate an overall average performance inferior to that of single-task learning models, such as LSTM. This disparity stems from the utilization of a hard parameter-sharing mechanism in these MTL models, wherein shared parameters are employed across multiple tasks, potentially leading to information confusion. Different tasks may possess unique feature representation requirements, and the adoption of shared parameters may impede the model’s ability to distinguish between the features of distinct tasks, consequently resulting in performance degradation or mutual interference among tasks. For example, in the MIMIC-III Infection dataset, although LSTM’s performance on the “Fever” task exhibits relative improvement compared to single-task learning, it noticeably deteriorates performance on the “Infection” and “Mortality” tasks. This pattern is similarly observed in the other two datasets.

To tackle this issue, we propose the introduction of a task-interdependent temporal feature-sharing mechanism that captures the dynamic correlations among tasks over time. Moreover, by utilizing GTAs for adaptive feature aggregation and enhanced sharing, our approach strengthens the learning of shared features that display strong correlations. The outcome is a clear improvement in the overall average performance of the model.

Based on the aforementioned comparative analysis results, we conducted an effective analysis of the cross-timestep feature-sharing mechanism based on GATs, which was proposed by us. Firstly, during the process of learning the graph structure, we established a sparse graph structure by selecting features that exhibit strong correlations among tasks. The underlying rationale behind this choice was that only a limited number of past observations are relevant to future predictions, necessitating the disregard of irrelevant timesteps. As a result, in the subsequent self-attention process, the softmax mapping no longer assigns scores to these timesteps, and the feature aggregation no longer incorporates features from these timesteps. This reduction in the inclusion of irrelevant timesteps not only diminishes the time required for learning the graph structure but also accelerates the attention process. Moreover, we opted to perform feature selection across timesteps prior to the softmax operation to prevent the loss of valuable features. To substantiate the effectiveness of our cross-temporal step feature-sharing strategy, we compared the predictive results of the model employing this sharing strategy (G-MTL) with those of the model based on our proposed strategy without feature sharing (Nonshared-MTL) using MIMIC-III Infection dataset. Figure 5 clearly illustrates that, in nearly all cases, G-MTL outperforms Non-shared MTL in terms of testing accuracy, thereby confirming the efficacy of our proposed cross-temporal step feature-sharing strategy.

### 4.6. Ablation Study

To assess the necessity of each component in our approach, we conducted experiments where we excluded each component individually to observe the degradation in model performance. Firstly, we replaced the learned graph with a static complete graph, connecting all nodes to the current node. This was done to investigate the importance of the graph structure. Secondly, we disabled the intra-task self-attention mechanism by assigning equal weights to all timestep features, meaning that each timestep feature contributed equally to the prediction result. Finally, we only considered feature sharing at the same timestep, eliminating the graph structure and the existence of attention-based feature sharing across timesteps between tasks. From Figure 6 and Table 5, it is evident that removing these components significantly lowers the prediction accuracy of the corresponding model (p<0.001, MANOVA) compared to G-MTL.

The results are summarized in Figure 7, Table 6, and provide the following findings:
Substituting the acquired graph structure with the complete graph led to a decline in prediction performance across all tasks. This observation serves as evidence that learning the graph structure enhances model performance, especially for datasets with temporal dependencies. The fundamental explanation behind this phenomenon can be attributed to the restricted relevance between a small subset of past observed outcomes and future predictions. As a result, we have diligently selected the most highly correlated timestep features among tasks, with the constraint of retaining only the top K, to construct a sparse graph.In our experiments, the exclusion of the intra-task self-attention mechanism exhibited the most detrimental impact on the model’s performance. This outcome arises from the fact that the correlations between features at distinct timesteps dynamically fluctuate and contribute disparately to the prediction of future outcomes. Treating all features at each timestep uniformly introduces noise and misleads the model. Hence, it becomes evident that relying solely on the dynamic dependencies between features across tasks within a multi-task learning framework is insufficient for accurate time series prediction.Furthermore, considering the variant model that only incorporates same-step feature sharing, it is evident that the model’s performance is significantly inferior to that of cross-timestep feature sharing. Cross-timestep feature sharing fully accounts for the correlation between different task features and enhances the learning of shared features associated with each task. This observation further validates the importance of the cross-timestep feature-sharing strategy.

### 4.7. Hyper-Parameter Analysis

We demonstrate the influence of hyperparameters on the performance of G-MTL. Figure 8 illustrates how the sparsity indicator of the graph structure (*TopK*) and the number of hidden layers greatly affect the forecasting precision for each task in the model. Furthermore, Table 7 quantitatively verifies that the accuracy of each task significantly differs (p<0.001, MANOVA) for different TopK values. Similar findings are observed for different hidden layers (Table 8, p<0.001, MANOVA). We verify the key hyper-parameters, i.e., the indicator of graph structure sparsity (TopK) and the number of hidden layers.

Therefore, it is crucial to carefully select the hyperparameters that best suit our model. Figure 9a presents the accuracy performance of our model on the MIMIC-III Infection dataset. In this plot, we test TopK in $\left\{10\%,15\%,20\%,25\%,30\%,35\%,40\%\right\}$. In multi-task learning, some tasks may underperform due to task imbalance, as depicted by the line graph in Figure 9a. The histogram highlights how the accuracy of all three tasks on the dataset improves or decreases to varying degrees with changes in TopK. For instance, the accuracy of Task 3 significantly improves at TopK=15%, while Task 1 shows a significant improvement at TopK=30%. However, both changes significantly compromise the accuracy of the corresponding other tasks, particularly Task 2. Considering the overall model performance, the task imbalance issue is alleviated when TopK is set to 20%. Figure 9b indicates that the number of hidden layers also affects the accuracy of different tasks. For optimal overall performance, the model performs better when the number of hidden layers is set to 2.

## 5. Conclusions

In this paper, we propose G-MTL, a graph-based multi-task time series prediction framework that effectively captures both global dynamic dependencies within tasks and local dynamic dependencies across timesteps between tasks. Although existing multi-task learning models enhance predictive performance by sharing features between tasks, the dependencies between task-specific features and shared features may vary across timesteps. Furthermore, the correlation of shared features with tasks also changes at different timesteps. To address these challenges and model the dynamic dependencies across timesteps and tasks, we propose a novel multi-task time series prediction framework for feature sharing across timesteps. This framework incorporates an adaptive learning approach to capture the dynamic dependencies between tasks using sparse graph structures. It explicitly models the correlation between features across different timesteps and tasks. Moreover, we introduce a flexible cross-timestep feature-sharing strategy based on graph attention to enhance the learning of features strongly correlated with each task. To evaluate the effectiveness of G-MTL, we conducted experiments on three commonly used datasets and demonstrated its superiority compared to existing models.

In summary, our work enriches research in multi-task time series forecasting in three key aspects: (1) A new multi-task time series prediction model that can simultaneously capture the dynamic dependencies of intra-task and inter-task features. (2) A novel method for capturing the dynamic feature dependencies between tasks through adaptive learning of sparse graph structures. It explicitly models the correlation between features across different timesteps and tasks. (3) A flexible cross-timestep feature-sharing strategy based on graph attention can enhance the learning of features strongly correlated to each task. In the future, we are committed to two more realistic scenarios, including unbalanced and interpretable shared learning among multiple tasks.

## Figures and Tables

**Figure 1 entropy-25-01136-f001:**
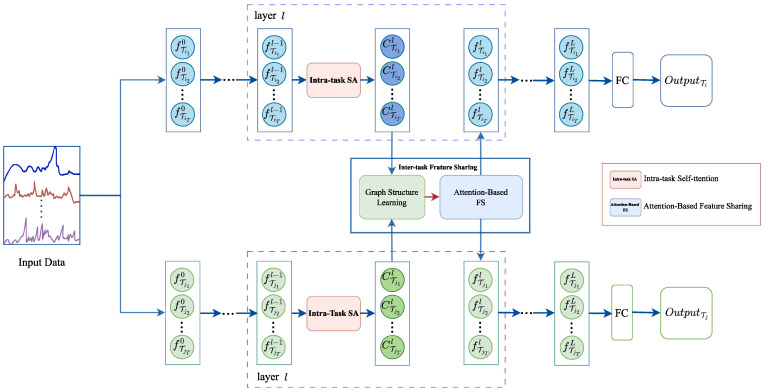
Model general framework diagram.

**Figure 2 entropy-25-01136-f002:**
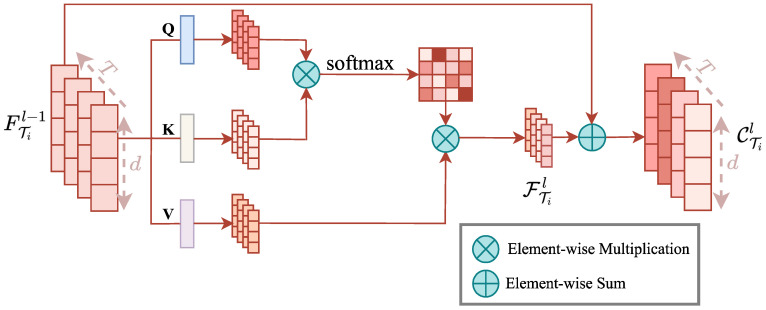
Intra-task feature sharing.

**Figure 3 entropy-25-01136-f003:**
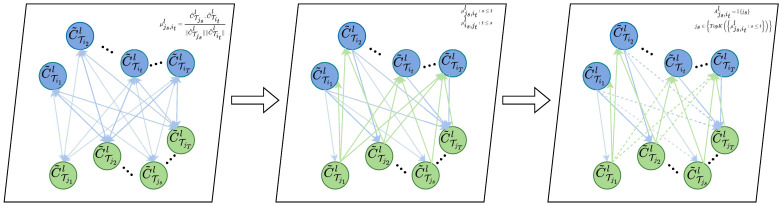
Graph structure learning.

**Figure 4 entropy-25-01136-f004:**
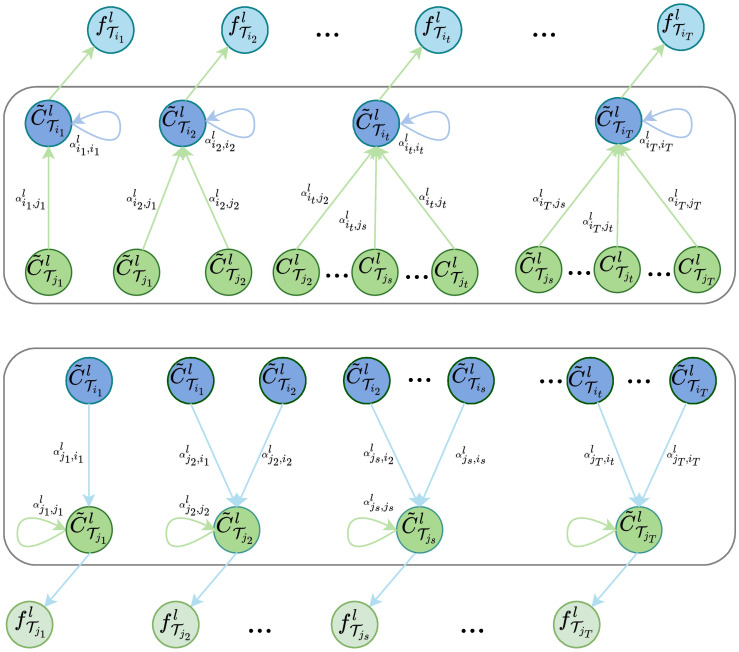
Inter-task feature sharing across timesteps.

**Figure 5 entropy-25-01136-f005:**
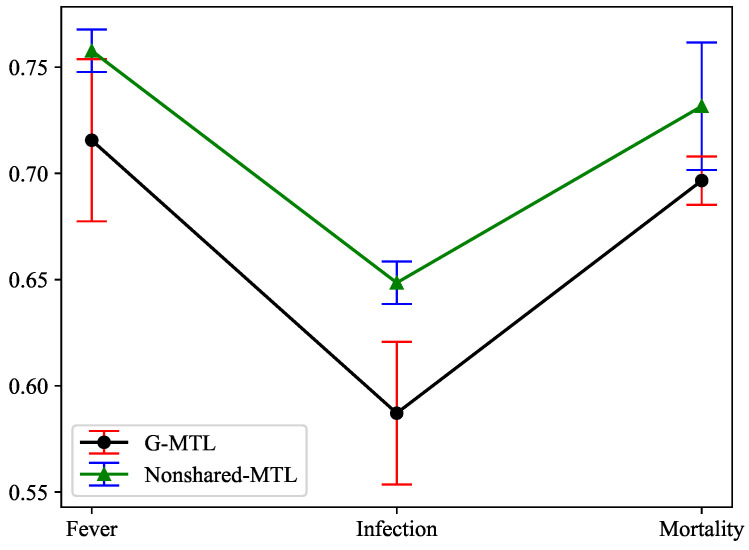
Illustration of the effectiveness analysis of a cross-timestep feature-sharing strategy.

**Figure 6 entropy-25-01136-f006:**
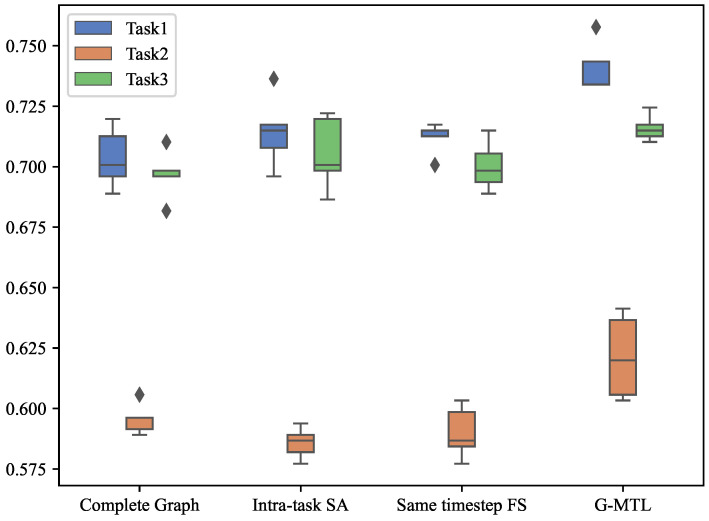
Comparison of the impact of each model component on model accuracy.

**Figure 7 entropy-25-01136-f007:**
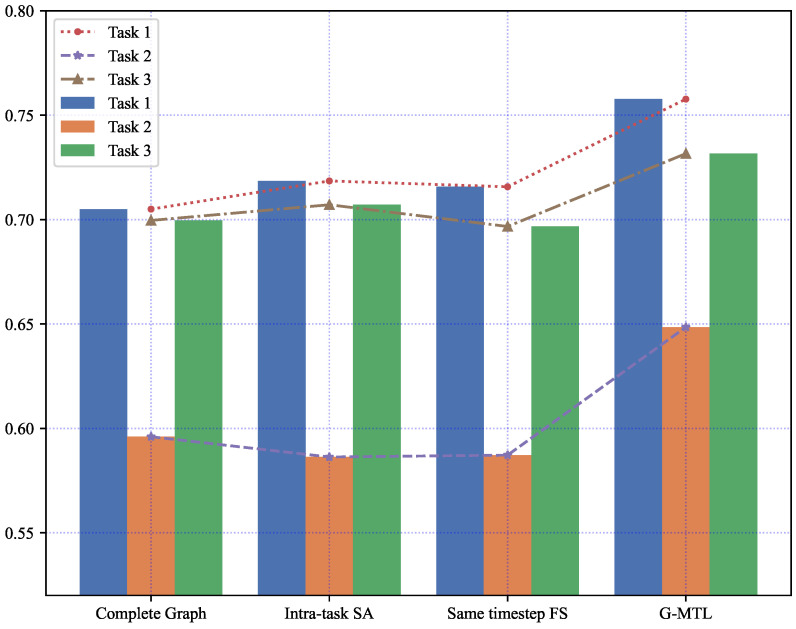
Ablation test results of G-MTL on MIMIC-III Infection dataset.

**Figure 8 entropy-25-01136-f008:**
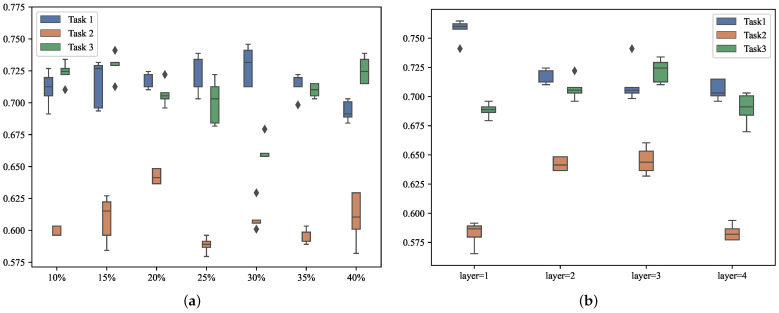
Illustration of the impact of hyper-parameters on the accuracy of each task on the MIMIC-III Infection dataset. (**a**) Comparison of the accuracy of each task on different *TopK*. (**b**) Comparison of the accuracy of each task on different hidden layers.

**Figure 9 entropy-25-01136-f009:**
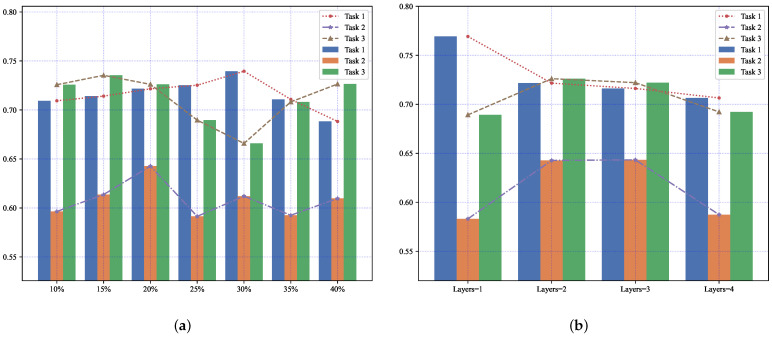
Each task forecasting accuracy of G-MTL with different hyper-parameters on MIMIC-III Infection dataset. (**a**) shows the effect of *TopK* on the accuracy of each task. (**b**) shows the effect of the number of hidden layers on the accuracy of each task.

**Table 1 entropy-25-01136-t001:** Summary of the used datasets.

Datasets	Samples	Dimension	Frequency	Training
MIMIC-III Infection	1921	12	hourly	1500
PhysioNet	4000	29	hourly	3000
MIMIC-III Heart Failure	3557	16	hourly	2500

**Table 2 entropy-25-01136-t002:** Task performance on the MIMIC-III Infection dataset. We report average AUROC and standard error over five runs, where the optimal performances are bold.

	Models	Fever	Infection	Mortality	Average
STL	LSTM	0.6738 ± 0.02	0.6860 ± 0.02	0.6373 ± 0.02	0.6657 ± 0.02
	Transformer	0.7110 ± 0.01	0.6500 ± 0.01	0.6766 ± 0.01	0.6792 ± 0.01
	RETAIN	0.6826 ± 0.01	0.6655 ± 0.01	0.6054 ± 0.02	0.6511 ± 0.01
	UA	0.6987 ± 0.02	0.6504 ± 0.02	0.6168 ± 0.05	0.6553 ± 0.02
	SAnD	0.6958 ± 0.02	0.6829 ± 0.01	0.7073 ± 0.02	0.6953 ± 0.01
	AdaCare	0.6354 ± 0.02	0.6256 ± 0.03	0.6217 ± 0.01	0.6275 ± 0.08
MTL	LSTM	0.7006 ± 0.03	0.6686 ± 0.02	0.6261 ± 0.03	0.6651 ± 0.02
	Transformer	0.7025 ± 0.01	0.6479 ± 0.02	0.6420 ± 0.02	0.6641 ± 0.02
	RETAIN	0.7059 ± 0.02	0.6635 ± 0.01	0.6198 ± 0.05	0.6630 ± 0.02
	UA	0.7124 ± 0.01	0.6489 ± 0.02	0.6325 ± 0.04	0.6646 ± 0.02
	SAnD	0.7041 ± 0.01	0.6818 ± 0.02	0.6880 ± 0.01	0.6913 ± 0.01
	AdaCare	0.5996 ± 0.01	0.6163 ± 0.02	0.6283 ± 0.01	0.6148 ± 0.00
	TP-AMTL	0.7081 ± 0.01	**0.7173 ± 0.01**	0.7112 ± 0.01	0.7112 ± 0.01
	G-MTL(ours)	**0.7577 ± 0.01**	0.6485 ± 0.01	**0.7316 ± 0.03**	**0.7126 ± 0.03**

**Table 3 entropy-25-01136-t003:** Task performance on the PhysioNet dataset. We report average AUROC and standard error over five runs, where the optimal performances are bold.

	Models	stay < 3	Cardiac	Recovery	Mortality	Average
STL	LSTM	0.7673 ± 0.09	0.9293 ± 0.01	0.8587 ± 0.01	0.7100 ± 0.01	0.8163 ± 0.03
	Transformer	0.8953 ± 0.01	0.9283 ± 0.02	0.8721 ± 0.01	0.6796 ± 0.02	0.8380 ± 0.01
	RETAIN	0.7407 ± 0.04	0.9236 ± 0.01	0.8148 ± 0.04	0.7080 ± 0.02	0.7968 ± 0.03
	UA	0.8556 ± 0.02	0.9335 ± 0.01	0.8712 ± 0.01	0.7283 ± 0.01	0.8471 ± 0.01
	SAnD	**0.8965 ± 0.02**	0.9369 ± 0.01	0.8838 ± 0.01	0.7330 ± 0.01	0.8626 ± 0.01
	AdaCare	0.7508 ± 0.06	0.8610 ± 0.01	0.7700 ± 0.03	0.6595 ± 0.02	0.7603 ± 0.07
MTL	LSTM	0.7418 ± 0.09	0.9233 ± 0.01	0.8472 ± 0.02	0.7228 ± 0.01	0.8088 ± 0.03
	Transformer	0.8532 ± 0.03	0.9291 ± 0.01	0.8770 ± 0.01	0.7358 ± 0.01	0.8488 ± 0.01
	RETAIN	0.7613 ± 0.03	0.9064 ± 0.01	0.8160 ± 0.04	0.6944 ± 0.03	0.7945 ± 0.03
	UA	0.8573 ± 0.03	0.9348 ± 0.01	0.8860 ± 0.01	0.7569 ± 0.02	0.8587 ± 0.02
	SAnD	0.8800 ± 0.03	0.9410 ± 0.00	0.8607 ± 0.01	0.7612 ± 0.02	0.8607 ± 0.06
	AdaCare	0.8746 ± 0.01	0.7211 ± 0.01	0.6348 ± 0.02	0.7457 ± 0.03	0.7440 ± 0.08
	AMTL-LSTM	0.7600 ± 0.08	0.9254 ± 0.01	0.8066 ± 0.01	0.7167 ± 0.01	0.8022 ± 0.03
	TP-AMTL	0.8953 ± 0.01	0.9416 ± 0.01	0.9016 ± 0.01	0.7586 ± 0.01	0.8743 ± 0.01
	G-MTL (ours)	0.8520 ± 0.01	**0.9780 ± 0.00**	**0.9300 ± 0.01**	**0.8480 ± 0.01**	**0.9020 ± 0.02**

**Table 4 entropy-25-01136-t004:** Task performance on the MIMIC-III Heart Failure dataset. We report average AUROC and standard error over five runs, where the optimal performances are bold.

	Models	Ischemic	Valvular	Heart Failure	Mortality	Average
STL	LSTM	0.7072 ± 0.01	0.7700 ± 0.02	0.6899 ± 0.02	0.7169 ± 0.03	0.7210 ± 0.01
	RETAIN	0.6573 ± 0.03	0.7875 ± 0.01	0.6850 ± 0.01	0.7027 ± 0.02	0.7081 ± 0.01
	UA	0.6843 ± 0.01	0.7728 ± 0.02	0.7090 ± 0.01	0.7191 ± 0.01	0.7213 ± 0.01
MTL	LSTM	0.6838 ± 0.02	0.7808 ± 0.02	0.6965 ± 0.01	0.7093 ± 0.02	0.7254 ± 0.02
	Transformer	0.6801 ± 0.01	0.7693 ± 0.01	0.7098 ± 0.02	0.7008 ± 0.02	0.7150 ± 0.02
	RETAIN	0.6649 ± 0.01	0.7532 ± 0.03	0.6868 ± 0.02	0.7023 ± 0.03	0.7018 ± 0.02
	UA	0.6917 ± 0.01	0.7868 ± 0.01	0.7073 ± 0.01	0.7029 ± 0.01	0.7222 ± 0.01
	AMTL-LSTM	0.6963 ± 0.01	**0.7997 ± 0.02**	0.7006 ± 0.01	0.7108 ± 0.01	0.7268 ± 0.01
	TP-AMTL	0.7113 ± 0.01	0.7979 ± 0.02	0.7103 ± 0.01	0.7185 ± 0.02	0.7345 ± 0.01
	G-MTL (ours)	**0.7666 ± 0.02**	0.7845 ± 0.01	**0.7271 ± 0.00**	**0.7540 ± 0.01**	**0.7581 ± 0.02**

**Table 5 entropy-25-01136-t005:** Multivariate Variance Analysis (MANOVA) of each component on MIMIC-III Infection dataset.

Models	Value	Num DF	Den DF	F Value	Pr > F
Wilks’ lambda	0.0825	9.0000	34.2229	6.7977	0.0000
Pillai’s trace	1.1296	9.0000	48.0000	3.2211	0.0039
Hotelling–Lawley trace	8.5662	9.0000	19.0526	12.7611	0.0000
Roy’s greatest root	8.2586	3.0000	16.0000	44.0461	0.0000

**Table 6 entropy-25-01136-t006:** Ablation test results of G-MTL on MIMIC-III Infection dataset, the optimal performances are bold.

Methods	Fever	Infection	Mortality
Complete Graph	0.7050 ± 0.01	0.5960 ± 0.01	0.6996 ± 0.01
Intra-Task SA ^1^	0.7185 ± 0.02	0.5863 ± 0.01	0.7071 ± 0.01
Same Timestep FS ^2^	0.7157 ± 0.04	0.5872 ± 0.03	0.6967 ± 0.01
G-MTL (ours)	**0.7577 ± 0.01**	**0.6485 ± 0.01**	**0.7316 ± 0.03**

^1^ Intra-task Self-Attention; ^2^ Same Timestep Feature-sharing.

**Table 7 entropy-25-01136-t007:** Multivariate Variance Analysis (MANOVA) of the number of *TopK* on MIMIC-III Infection dataset.

*TopK*	Value	Num DF	Den DF	F Value	Pr > F
Wilks’ lambda	0.0385	18.0000	74.0244	8.8931	0.0000
Pillai’s trace	1.7761	18.0000	84.0000	6.7720	0.0000
Hotelling–Lawley trace	7.5641	18.0000	18.0000	10.5407	0.0000
Roy’s greatest root	4.9715	6.0000	28.0000	23.2002	0.0000

**Table 8 entropy-25-01136-t008:** Multivariate Variance Analysis (MANOVA) of the number of hidden layers on MIMIC-III Infection dataset.

Hidden Layers	Value	Num DF	Den DF	F Value	Pr > F
Wilks’ lambda	0.0079	9.0000	34.2229	23.9178	0.0000
Pillai’s trace	1.9403	9.0000	48.0000	9.7658	0.0000
Hotelling–Lawley trace	23.3231	9.0000	19.0526	34.7447	0.0000
Roy’s greatest root	18.9611	3.0000	16.0000	101.1259	0.0000

## Data Availability

Not applicable.

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
