# Peer review of "Multi-Task Time Series Forecasting Based on Graph Neural Networks"

_entropy, 2023, doi:10.3390/e25081136_

Round 1

Reviewer 1 Report

1. Comparative analysis is not a sufficient justification to be published.

2. It is required to justify adequately the main contribution of the paper in the context of Entropy Journal.

3. It is mandatory to adequately justify the selection of used data. Besides data should be analyzed before to be used.

4. Finally, it is required to make available used data and include all the implementation parameters in order that readers can reproduce proposed results.

5. It should be great if authors can include a repository of used codes.

This paper is well written and organized, there are only minor tyoing errors.

Reviewer 2 Report

The paper is well-written, well organized, and it presents an interesting model.

For your final version of the paper:

1) I didn't get what is STL that you referred to in your model comparisons.

 Is it Seasonal-Trend decomposition using LOESS? If it is I recommend a short explanation of that. I recommend this reference that covers this topic: Aggregating Prophet and Seasonal Trend Decomposition for Time Series Forecasting of Italian Electricity Spot Prices.

Considering comment 1, I think you should check all your used acronyms.

2) There are some typos, like in lines 397, 389, 381, 214, 222, 39 and so on. The majority of these typos are regarding the missing spaces.

3) There are paragraphs too long, like subsection 2.3 and the conclusion section.

In general, the paper sounds good.

Many typos.

Round 2

Reviewer 1 Report

I am still considering that authors should improve definition of paper main contribution.

I find several typing and grammatical errors that should be corrected.
